# T Cell Responses during Human Immunodeficiency Virus/*Mycobacterium tuberculosis* Coinfection

**DOI:** 10.3390/vaccines12080901

**Published:** 2024-08-09

**Authors:** José Alejandro Bohórquez, Chinnaswamy Jagannath, Huanbin Xu, Xiaolei Wang, Guohua Yi

**Affiliations:** 1Department of Cellular and Molecular Biology, The University of Texas Health Science Center at Tyler, Tyler, TX 75708, USA; josealejandro.bohorquezgarzon@uttyler.edu; 2Center for Biomedical Research, The University of Texas Health Science Center at Tyler, Tyler, TX 75708, USA; 3Department of Medicine, The University of Texas at Tyler School of Medicine, Tyler, TX 75708, USA; 4Department of Pathology and Genomic Medicine, Center for Infectious Diseases and Translational Medicine, Houston Methodist Research Institute, Houston, TX 77030, USA; cjagannath@houstonmethodist.org; 5Tulane National Primate Research Center, Tulane University School of Medicine, Tulane University, Covington, LA 70112, USA; hxu@tulane.edu (H.X.); xwang@tulane.edu (X.W.)

**Keywords:** tuberculosis, HIV, coinfection, T cell response, granuloma

## Abstract

Coinfection with *Mycobacterium tuberculosis* (*Mtb*) and the human immunodeficiency virus (HIV) is a significant public health concern. Individuals infected with *Mtb* who acquire HIV are approximately 16 times more likely to develop active tuberculosis. T cells play an important role as both targets for HIV infection and mediators of the immune response against both pathogens. This review aims to synthesize the current literature and provide insights into the effects of HIV/*Mtb* coinfection on T cell populations and their contributions to immunity. Evidence from multiple in vitro and in vivo studies demonstrates that T helper responses are severely compromised during coinfection, leading to impaired cytotoxic responses. Moreover, HIV’s targeting of *Mtb*-specific cells, including those within granulomas, offers an explanation for the severe progression of the disease. Herein, we discuss the patterns of differentiation, exhaustion, and transcriptomic changes in T cells during coinfection, as well as the metabolic adaptations that are necessary for T cell maintenance and functionality. This review highlights the interconnectedness of the immune response and the pathogenesis of HIV/*Mtb* coinfection.

## 1. Introduction

*Mycobacterium tuberculosis* (*Mtb*), the causative agent of tuberculosis (TB), remains one of the most devastating pathogens in public health, ranking as the second leading infectious disease killer in 2022 [1]. This high mortality is related to the widespread nature of *Mtb* infection, with approximately one-quarter of the global population infected. Although infection is usually contained by the immune system, 5–10% of those infected with *Mtb* develop active TB, and the majority develop latent *Mtb* infection (LTBI) [1,2]. During LTBI, the pathogen modulates the host immune response to survive and evade host immunosurveillance within granuloma structures, a hallmark of this disease state [3]. These structures are formed early after *Mtb* infection, by continued recruitment and aggregation of immune cells, after the initial contact with the pathogen. Macrophages are essential for granuloma formation, as they form the inner layers and serve as the foundation for the granuloma. Multiple myeloid and lymphoid cell subsets, as well as non-hematopoietic cells, form the outer layers of the structure and help in keeping the pathogen latent at its core. The formation of the granuloma and establishment of the LTBI disease form involves careful modulation of the host’s immunity by *Mtb*.

The *Mtb* pathogenesis during LTBI can be altered by conditions such as malnutrition, diabetes, or immunosuppressive infections, leading to the progression from latent to active TB, which is fatal if left untreated [4,5,6]. In this regard, the human immunodeficiency virus (HIV) significantly contributes to the development of active TB [7,8,9]. It is estimated that LTBI individuals who are coinfected with HIV are 16 times more likely to develop active TB compared to those without HIV infection [1]. HIV primarily targets immune cell populations (mainly CD4^+^ T cells), leading to a dysregulation of the immune response that facilitates the proliferation of opportunistic infections, as well as the activation of latent infections [10,11]. Among these opportunistic infections, pathogens in the *Mycobacterium tuberculosis* complex (MTBC), including *Mtb*, the most prevalent of them, are particularly important causes of disease in people living with HIV (PLWH). Moreover, T cell populations are present in the granuloma structures that compartmentalize *Mtb* during latent infection [3]. HIV-induced dysregulation of these T cells can lead to the disorganization of the granuloma structure, allowing the pathogen to be released into the lungs, where it can proliferate freely in the absence of an adequate immune response [12]. In particular, a synergistic relation between CD4^+^ cells and macrophages is necessary for control of *Mtb*, and its alteration has an effect on the cell composition within the granuloma. This is reflected in studies reporting a decrease in the giant multinucleated cell population in granulomas from coinfected patients [13,14].

Coinfection plays a major role in the pathogenesis of both *Mtb* and HIV, which have mainly been reviewed from a microbiological and clinical standpoint. Similarly, the epidemiological importance of both diseases and their synergistic effects have been evaluated as well. However, despite the importance of T cells in the immune response, the alterations occurring in these cell populations during HIV/*Mtb* coinfection have been approached as part of these works, rather than a sole, in-depth focus of a literature review. The present work aims to summarize the current knowledge regarding the effect of HIV/*Mtb* coinfection on T cell populations. This review includes results obtained in studies using both targeted and high-throughput multi-omics methodologies, encompassing multiple aspects, including a general overview of phenotypic changes and more specific functional alterations in these cells. Finally, we address the modulation of transcriptomic and metabolic pathways within T cell populations resulting from HIV/*Mtb* coinfection.

## 2. Phenotypic Alterations in T Cells during HIV/*Mtb* Coinfection

### 2.1. CD4^+^ T Cells

As the primary target of HIV infection, CD4^+^ T cells are severely affected during HIV/*Mtb* coinfection. While *Mtb* infection has been shown to increase the percentage of CD4^+^ T cells in PBMCs, the hallmark of HIV infection is a severe depletion of this cell subset. During HIV/*Mtb* coinfection, a massive decrease in the relative and absolute number of CD4^+^ T cells is evidenced in PBMCs [15]. Interestingly, the depletion of CD4^+^ cells is found to occur earlier and preferentially in tissues, including the lungs [16]. The gastrointestinal tract is also a notably major site of severe CD4^+^ T cell depletion following HIV and simian immunodeficiency virus (SIV) infection [17,18,19]. This tissue-specific depletion may explain the increased risk of active TB in PLWH, with normal CD4^+^ T cell counts in PBMCs [20].

Notably, *Mtb*-specific CD4^+^ T cells are more susceptible to depletion by HIV/SIV than naïve cells, leading to a specific decrease at the site of *Mtb* infection [21,22,23]. This increased susceptibility is likely the result of the upregulation of CCR5 in CD4^+^ T cells and other immune cell types following *Mtb* infection [24,25]. CCR5 expression aids in cell migration to the site of infection and is part of the normal immune response against *Mtb* infection; however, it also has a role as the primary coreceptor for HIV [26], and its upregulation is bound to facilitate CD4^+^ cell depletion by allowing for HIV infection of these cells. Recently, Foreman et al. [20] used an Indian-origin rhesus macaque model to demonstrate that the depletion of CD4^+^ T cells within granulomas occurs early after coinfection with SIV, preceding alterations detected in PBMCs or lymphoid organs. This is likely to have a direct detrimental effect on the control of the *Mtb* inside the granuloma structure, as CD4^+^ cells have been found to be necessary for the formation of these structures [27]. Additionally, the interaction of CD4^+^ cells with other immune cell types within the granuloma is affected after coinfection with HIV, leading to structural changes and contributing to the failure in *Mtb* control.

Although *Mtb*-specific cells have been proven to be more susceptible to HIV coinfection, certain *Mtb*-related microenvironments have also been reported to reduce the capacity of HIV to replicate and integrate in CD4^+^ cells, even if viral entry remains successful. A recent study that examined the effect of pleural effusion from active TB patients on CD4^+^ cells revealed that reverse transcription and genomic integration were reduced in this microenvironment [28]. Therefore, even though HIV coinfection facilitates the activation of *Mtb*, the effect of *Mtb* on the reactivation of latent HIV, or even the establishment of viral latency itself, is not as straightforward, at least in certain microenvironments.

The depletion of CD4^+^ T cells in PBMCs of PLWH, although a hallmark of HIV infection, can be transient and reversible with anti-retroviral therapy (ART). The replenishment of CD4^+^ cells in PBMCs following ART is associated with early differentiated cells (CD4+CD27+CD45RO+), which contribute to Th1, Th17, and Th22 responses to *Mtb* [29,30]. However, in active TB cases, the CD4^+^ cell population shifts toward a more differentiated phenotype (CD27-), regardless of HIV status [31]. Late-differentiated CD4^+^ T cells remain depleted in PLWH following cART [29], hindering an effective immune response against *Mtb* in active TB patients who are coinfected with HIV.

Another phenotypical aspect of CD4^+^ memory T cells that are implicated in immune responses against *Mtb* is the expression of different chemokine receptors on their surface [30]. The depletion of *Mtb*-specific CD4^+^ T cells during HIV coinfection leaves behind less differentiated phenotypes with reduced motility [20,32,33]. This compromises bacterial infection control and likely contributes to disease progression.

### 2.2. CD8^+^ T Cells

CD8^+^ T cells play important roles in both *Mtb* and HIV infections. HIV-infected patients develop robust, virus-specific immune responses, although these fail to control the infection [34,35,36]. During *Mtb* infection, CD8^+^ lymphocytes are recruited to infection sites and are present in the granuloma structures [3,37]. Here, this immune cell subset appears to play a major role during latency, while CD4^+^ T cells are crucial during acute infection [38,39].

HIV/*Mtb* coinfection alters the CD8^+^ T cell phenotypes, increasing their differentiation and reducing the proportion of naïve cells. This is reflected in the marked expansion of effector memory (EM; CD45RA- CD27-) and terminal effector (TE; CD45RA+ CD27-) CD8^+^ T cell subsets in PBMCs from coinfected active TB patients [40,41]. However, these TE cells are not fully differentiated, exhibiting low CD45RA expression, which likely impairs their responsiveness. Furthermore, during HIV/*Mtb* coinfection, some CD8^+^ T cell populations express the PD-1 surface marker, which might indicate poor functionality and exhaustion in these cells [7,40]. This might be related to CD4^+^ T cell depletion, given that these cells have been shown to help prevent CD8^+^ T cell exhaustion [42].

CD8^+^ T cell phenotypes in the PBMCs of HIV/*Mtb*-coinfected patients change following treatment with cART, although these changes vary depending on TB status [41]. HIV/LTBI patients show an increase in the proportion of less differentiated, naïve CD8^+^ T cells, similar to CD4^+^ T cells, while HIV patients with active TB show a slight increase in effector subsets.

Alterations in CD8^+^ T cell populations during HIV/*Mtb* coinfection are less widely reported in non-blood samples relative to CD4^+^ T cells. However, NHP models show an increase in the proportion of CD8^+^ T cells in bronchoalveolar lavage (BAL) following SIV infection, which reverts after cART. Coinfection of SIV-infected, cART-treated NHPs with *Mtb* leads to a further decrease in this cell subset [43]. CD8^+^ T cells in lymph nodes from SIV-infected NHPs exhibit intermediate effector phenotypes, mirroring the changes evidenced in PBMCs. However, similar changes during SIV/*Mtb* coinfection have not been confirmed [44].

### 2.3. Unconventional T Cell Subsets

Other T cell subsets, such as mucosal-associated invariant T cells (MAIT), also play roles in immune responses against HIV and *Mtb*. MAIT cells delay initial CD4^+^ cell priming post-*Mtb* infection and decrease in circulation in active TB patients and PLWH [45,46,47,48,49]. This decrease is attributed to recruitment to the affected tissues. However, HIV/*Mtb* coinfection studies show variable effects on the blood’s MAIT cell percentages [50,51,52]. This is at least partially explained by the existence of multiple phenotypes of MAIT cells; CD4^+^ MAIT cells are severely depleted, while remaining phenotypes appear unaltered due to overall CD4^+^ T cell depletion. Moreover, MAIT cells from HIV/*Mtb*-coinfected patients express PD-1 and show altered chemokine receptor expression [46,52]. Findings using an NHP SIV/*Mtb* model suggest that MAIT cells migrate to the lung rather than lymph node during coinfection but are limited in their abundance and functionality in the granuloma [53,54].

Gamma delta T cells (γδ T cells) also play roles in bacterial and viral infections [55,56]. While multiple phenotypes of γδ T cells can be found in the blood, single infection with HIV or *Mtb* decreases γδ T cell proportions in PBMCs, especially in the Vδ2 phenotype [57,58]. Similarly, HIV/*Mtb*-coinfected patients show an overall depletion in γδ T cells, particularly in Vδ2 and Vγ2δ2 subsets, with increased Vδ1 T cells, typically found in tissues [59]. However, surviving Vδ2 subsets exhibit increased CD4^+^CD8^+^ T cells, indicating a substantial change in this cell population, which is typically CD4-CD8-, potentially altering their functionality and capacity to lyse antigens.

HIV/*Mtb* coinfection also alters the proportion and phenotype of invariant natural killer T cells (iNKT cells) [60,61], which rely on lipid antigen presentation, such as *Mtb* membrane lipids, for activation [62]. Alterations in the percentage of iNKT cells in PBMCs from HIV/*Mtb*-coinfected patients align with the overall depletion observed during HIV infection. Namely, the CD4^+^CD8- subset decreases, leading to a relative increase in CD4-CD8^+^ and CD4-CD8- phenotypes [35]. Furthermore, these cells exhibit increased CD107a expression, which is indicative of cytotoxic degranulation, particularly in cases of extrapulmonary TB. This study found an association between the recovery of iNKT cell percentages and the presentation of TB immune reconstitution inflammatory syndrome (TB-IRIS) after starting cART. This syndrome involves hyperinflammation and an exacerbated immune response against *Mtb*, leading to tissue damage and a worsened clinical picture after treatment [63]. Although the association between an iNKT cell increase and TB-IRIS is evident, the precise mechanism underlying this association remains unclear.

Another T cell population that is suggested to play a role during TB-IRIS is regulatory T cells (Treg) [64]. This immunosuppressive subset is relatively increased in HIV/*Mtb*-coinfected patients, likely due to the depletion of other CD4^+^ T cell subsets, as the absolute numbers of Treg cells remain unaffected [65]. Another report focusing on phenotypic changes within Treg populations identified two distinct phenotypes based on the CD25^+^ surface marker, exhibiting different patterns [66,67]. Unconventional Tregs (uTreg, CD4+CD25-FoxP3+) are elevated in PBMCs from HIV/*Mtb*-coinfected cohorts, with a relative reduction in conventional Tregs (cTreg, CD4+CD25+FoxP3+). In addition, uTreg cells display more differentiated profiles in terms of effector and memory markers, as well as the PD-1 surface marker of exhaustion. Although the exacerbated immune response in TB-IRIS has been proposed to be related to a sudden increase in helper T cell type I cell populations compared to Tregs, no causal association has been found [68].

## 3. Alterations in T Cell Functionality

### 3.1. Th1 T Cell Response

Th1 cytokines, such as interferon gamma (IFN-γ), tumor necrosis factor alpha (TNF-α), and interleukin 2 (IL-2), play crucial roles in the immune response against *Mtb*. The progression to active TB in LTBI patients following HIV infection has been associated with the early depletion of *Mtb*-specific Th1 cells [69]. However, the immune modulation during HIV/*Mtb* coinfection is complex, with variations depending on factors such as the cell phenotype, stimulation status, and type of stimuli used [70,71]. The absolute number of IFN-γ and TNF-α-producing CD4^+^ T cells is severely depleted in HIV/*Mtb*-coinfected patients [72]. This depletion may be a side effect of the overall CD4^+^ T cell depletion, as the proportion of cytokine-producing cells relative to the surviving population is not always affected. In fact, after stimulation with certain *Mtb* antigens, PBMCs from HIV/*Mtb*-coinfected cohorts exhibit polyfunctional CD4^+^ T cells that produce combinations of IFN-γ, TNF-α, and IL-2 [73,74].

Multiple studies have corroborated the presence of polyfunctional *Mtb*-specific CD4^+^ T cells in PBMCs from PLWH in an antigen-dependent manner [71,74,75,76]. However, in BAL, both the number and functionality of immune cells, including polyfunctional cells, are severely decreased in HIV/*Mtb*-coinfected individuals compared to those with HIV alone [16].

IL-2 production by CD4^+^ and CD8^+^ cells is reduced in HIV/*Mtb*-coinfected individuals [7,71]. Interestingly, the reconstitution of Th1 cell subsets following ART does not uniformly affect all phenotypes; IFN-γ-producing cells are replenished, while IL-2-producing cells remain depleted [77]. This discrepancy may be due to differences in the reconstitution of effector and memory T cell phenotypes, as described previously.

Unlike the variable responses of CD4^+^ T cells, the functionality of CD8^+^ T cells during HIV/*Mtb* coinfection is more clearly understood. CD8^+^ T cells producing Th1 cytokines are reduced in number, indicating diminished functionality, possibly due to poor stimulation from the depleted CD4^+^ subset [40,71,72,78]. Additionally, CD8^+^ T cells in HIV/*Mtb*-coinfected patients express the PD-1 surface marker, suggesting immune exhaustion [40]. Unconventional T cell phenotypes, such as γδ and MAIT cells, are also functionally impaired during HIV/*Mtb* coinfection, showing lower TNF-α and IFN-γ expression, respectively [50,79]. The functional impairment in MAIT cells can be reversed with ART.

### 3.2. Th2 T Cell Response

HIV infection may induce a switch from a Th1 to a Th2 immune response, characterized by increased IL-4 and IL-10 synthesis [79,80], potentially facilitating opportunistic infections, such as *Mtb*. Recent evidence suggests a more nuanced immunological landscape regarding the Th1/Th2 switch. While IL-4 levels are elevated in the plasma of both HIV and *Mtb* patients, they vary in coinfected individuals [81,82,83]. In addition, the number of IL-4-producing CD4^+^ T cells decreases in PBMCs during HIV/*Mtb* coinfection [7], which is linked to reduced expression of Gata-3 in memory CD4^+^ T cells. Another study reported increased plasma IL-4 concentrations in HIV/*Mtb*-coinfected patients after ART [84], although this increase was only evidenced in patients with higher pre-treatment CD4^+^ T cell counts.

IL-10, another major Th2 response cytokine, has been extensively studied in HIV and *Mtb* contexts [85,86]. However, the involvement of T cells in IL-10 production during HIV/*Mtb* coinfection is less characterized. Studies show increased plasma concentrations of IL-10 in coinfected individuals [81,84,87], with certain genotypes being associated with higher IL-10 production after HIV infection and a greater predisposition to coinfection [88]. Barham et al. [75] showed that IL-10 production of CD4^+^ T cell cultures in coinfected individuals correlated with cell activation markers and other Th1 and Th22 cytokines. However, whole PBMC cultures did not show differences in IL-10 concentration between HIV+ and HIV- individuals when stimulated with *Mtb* antigens.

### 3.3. Th17 T Cell Response

Given the important role of specific CD4^+^ T cell subsets in IL-17 production, it follows that the Th17 response is hindered during HIV/*Mtb* coinfection due to severe T cell depletion [89]. The *Mtb*-specific Th17 response is decreased in PBMCs from PLWH [71], and reports show that Th17 cells remain reduced even after treatment with ART or anti-TB drugs [90]. Interestingly, other cell populations increase their production of IL-17 during HIV/*Mtb* coinfection. Specifically, CD8^+^ and γδ T cells produce significantly higher levels of this cytokine in coinfected individuals [59,72].

For CD8^+^ lymphocytes, this increase is likely a response to elevated levels of tumoral growth factor beta (TGF-β), a known stimulant for IL-17 production [91], which increases after coinfection [72]. Th17 activation in γδ T cells may be linked to immune stimulation by *Mtb*, as shown in a mouse animal model where most IL-17 production after *Mtb* infection came from γδ T cells that were stimulated by IL-23 secreted by dendritic cells [92].

### 3.4. Th22 T Cell Response

As part of the immune response against *Mtb*, the Th22 T cell response, defined by IL-22 production, has been the focus of studies examining its modification during HIV coinfection [93,94]. Similar to Th17, a substantial portion of the Th22 response relies on CD4^+^ T cells, as these subsets produce IL-22 [95]. Remarkably, *Mtb*-specific CD4^+^ T cells have been identified as being the almost exclusive producers of IL-22 during *Mtb* infection in the absence of HIV [96]. Consequently, the Th22 T cell response is severely depleted during HIV/*Mtb* coinfection, reflected by decreased cytokine concentrations and a reduced number of IL-22-producing cells [30,96,97]. This impaired Th22 response is particularly evident in cases of active TB [96], although it has also been observed in LTBI patients who are coinfected with HIV [30].

### 3.5. Cytotoxic Granule Alterations in T Cells

Cytotoxic cytokines, such as perforin and granzyme A, are important components of T cell responses against pathogens. During HIV/*Mtb* coinfection, perforin production does not appear to be altered in either CD4^+^ or CD8^+^ T cells, despite its increase in HIV monoinfection [72]. However, γδ T cells show reduced perforin production in PBMCs from coinfected patients [59]. Intriguingly, the proportion of CD4^+^ T cells expressing granzyme A is significantly increased in the same cohort, as is the case for the CD8^+^ T cell subset. This indicates that the functionality of the remaining CD4^+^ T cells in HIV infection is likely stimulated by the *Mtb* antigen, implying a synergy between both antigens.

The alterations of T cell phenotypes and functionality are summarized in Figure 1.

## 4. Transcriptomic Alterations in T Cells during HIV/*Mtb* Coinfection

The emergence of techniques for transcriptomic regulation analysis has provided unprecedented information detailing immune response modulation [98]. These techniques have been extensively used to assess changes in gene expression during HIV and *Mtb* single infections [99,100,101,102]. In the case of HIV/*Mtb* coinfection, studies have focused on characterizing the immune cell transcriptome at systemic and local levels. A report by Guo et al. [103] compared PBMCs from HIV-infected and HIV/*Mtb*-coinfected individuals with those from a healthy cohort, revealing expected changes, such as a decrease in CD4^+^ T cells. A more in-depth analysis of different cell types in HIV/*Mtb*-coinfected individuals showed a wide array of CD8^+^ T cells (divided into eight different clusters), and only two clusters of CD4^+^ T cells were evidenced (naïve and Treg). Interestingly, the naïve CD4^+^ T cell population was higher in coinfected patients compared to the HIV single infection group, while the opposite pattern was observed in Tregs. Transcriptomic modification of Th1 immune response against *Mtb* in PLWH has been attributed to histone modifications in infected CD4^+^ T cells [104]. This leads to decreased production of macrophage-recruiting cytokines and a dysregulated transcription network in this cell subset, which appeared to be irreversible.

A meta-analysis of transcriptomic data collected from the PBMCs of HIV-infected and HIV/*Mtb*-coinfected individuals, assessed through microarray, identified 142 upregulated and 151 downregulated genes in the coinfected cohort [105]. Interestingly, although both groups were infected with HIV, the gene ontology enrichment suggested that the main pathways upregulated during coinfection were those involved in viral immunity and IFN stimulation. This approach, while not identifying the specific populations that are responsible for the upregulation of these pathways, implied the involvement of T cell populations. This is exemplified by the identification of upregulated genes in the T cell receptor signaling pathway.

At a local level, the analysis of BAL cells from HIV/*Mtb*-coinfected individuals showed a more balanced distribution of T cell subsets compared to PBMCs, with three clusters of CD4^+^ cells vs. four clusters of CD8^+^ cells [106]. However, the CD4^+^ cell subset was depleted compared to the *Mtb* single infection group. Remarkably, the functionality of the surviving T cells in BAL did not appear to be affected, contrary to previous reports [16]. This was evidenced by the upregulation of IFN-stimulated genes in all T cell phenotypes evaluated. Additionally, gene ontology analysis showed an upregulation of genes in multiple immune pathways, mainly those involved in the viral immune response and antigen processing and presentation. These immune-stimulated pathways were upregulated across multiple NK and T cell phenotypes.

## 5. Metabolic Alterations in T Cells during HIV/*Mtb* Coinfection

Although there have only been a limited number of studies on the metabolic effects of HIV/*Mtb* coinfection, and none exclusively focused on T cell metabolism, some abnormalities found in the plasma of coinfected patients are likely influenced by alterations in T cell subsets.

These studies primarily show an increase in energetic needs, leading to dysregulated energy uptake and a dependence on alternative energy sources. This is evidenced by significant changes in various metabolite groups during infection that are compatible with inflammation and increased catabolism [107]. The metabolism changes likely depend on the infection stage, as different T cell subsets use different energy sources: Th1, Th2, and Th17 cells favor glycolysis, whereas Tregs and memory T cells (both CD4^+^ or CD8^+^) mainly use lipid oxidation [108]. Exhausted T cells, which are prevalent during HIV/*Mtb* coinfection, also downregulate glycolytic genes and favor lipid oxidation [109]. In addition, Mtb infection has been associated with insulin resistance [110], which would be detrimental to cell populations that rely on glycolysis for energy production. In turn, this would favor the survival of exhausted and naïve cells, which are not able to mount an effective immune response.

The lipid metabolism is severely affected during HIV/*Mtb* coinfection, with studies showing a broad decrease in multiple lipid groups, including total cholesterol (TC), triglyceride (TG), low-density lipoprotein cholesterol (LDL-C), and high-density lipoprotein cholesterol (HDL-C) [107,111]. Specific alterations in long-chain fatty acids and their transport metabolites have been reported in both humans and humanized mouse models of coinfection [107,112]. These abnormalities suggest an impact on the oxidative phosphorylation pathway, which hinders cell activity. Moreover, some reports link fatty acid alterations to the HIV or *Mtb* pathogen load and immunological parameters, such as low CD4 counts [107,113,114]. Besides their function as an energy source, fatty acids are important for cell membrane turnover, and decreased concentrations may affect new membrane formation during cell proliferation. Interestingly, lipid-related metabolite decreases are worse in HIV/*Mtb*-coinfected individuals than in TB patients, suggesting a significant role of HIV in this dysregulation [107].

Abnormalities in the protein and amino acid metabolism are widely reported in HIV/*Mtb* coinfection. The main finding is an increased catabolism and lower net protein balance [115]. In terms of the T cell metabolism, alterations in tryptophan and related compounds are particularly relevant [116,117,118]. T lymphocyte activity is influenced by tryptophan levels [119,120], and abnormalities in its metabolism are reported in both HIV and *Mtb* single infections [121]. During HIV/*Mtb* coinfection, the indoleamine 2,3-dioxygenase 1 (IDO-1) concentration increases, indicating tryptophan degradation. This leads to higher levels of tryptophan subproducts in the plasma, although these subproducts are low in advanced disease stages [107,118]. Other amino acids, such as proline and phenylalanine, also show decreased levels in the plasma of coinfected patients [107] and play important roles in CD8^+^ T cell metabolism. Proline-rich tyrosine kinase-2 is necessary for CD8^+^ activation via the T cell receptor, while phenylalanine transport significantly increases in CD8^+^ T cells after activation [122,123].

## 6. Discussion

Understanding the dynamics between infection and immune response is crucial for developing host-directed intervention strategies [124,125]. In the case of HIV/*Mtb* coinfection, the involvement of two pathogens that modulate the immune response further complicates this understanding [10]. The recent focus on multi-omics approaches for this and other infections has generated an unprecedented volume of information, enabling a more thorough characterization of immune cell populations and their activation or suppression patterns in both healthy and disease states. Analyzing and presenting these data can be challenging, but it provides a more detailed picture of cellular and systemic alterations. In this review, we attempted to reconcile information from traditional and new-generation approaches regarding T cell responses during HIV/*Mtb* coinfection.

The numerous individual alterations in T cell response during coinfection are best understood as part of a network, in which any specific changes can affect the whole system and are enhanced or worsened by multiple factors. For example, while it is easy to attribute immunological changes during coinfection mainly to HIV, given that the most important immunological consequence of coinfection is the depletion of CD4^+^ T cells, it is actually *Mtb* infection that leads to an increase in CCR5 expression in CD4^+^ T cells. This, in turn, enhances the susceptibility of these cells to HIV, worsening an already deleterious effect [24,25]. Similarly, although metabolic changes during HIV/*Mtb* greatly overlap with those seen in single *Mtb* infection, HIV coinfection significantly contributes to the disruption of the lipid metabolism [107]. Therefore, it is crucial to keep the systemic perspective in mind when interpreting future studies that assess individual aspects of the immune response during HIV/*Mtb* coinfection, particularly those using high-throughput multi-omics techniques.

Moreover, assessing the effects of coinfection should also consider the multiple scenarios in which it occurs, as different compartments (e.g., blood vs. lung) exhibit somewhat different mechanisms of modulation [7,40]. While the compartmentalization of the immune response may seem at odds with the systemic interpretation of changes favored in previous studies, systemic alterations can explain localized changes. The differences between local and systemic alterations can be attributed to the availability of cell subsets in tissues, given the changes in migration capabilities that have been observed in the blood compartment and the suggested functional changes during coinfection [20]. In this regard, the use of novel technologies, such as spatial transcriptomics, may provide new insights into the pathogenesis of diseases that rely heavily on immunological and structural changes in the host. The significance of these techniques will be enhanced by experimental design, as the lack of biological controls (comparison with healthy subjects) was evident in the RNAseq-based transcriptomics studies included in this review [103,106]. This restricted their impact and the scope of comparisons between coinfection and single infection. Transcriptomic techniques are poised to be a crucial tool for studying infectious diseases in the years to come, and the gaps in knowledge for HIV/*Mtb* coinfection have been previously discussed [126].

Similarly, metabolomics studies aimed specifically at T cells can greatly advance our understanding of the immune response in HIV/*Mtb* coinfection, as the current evidence only allows for explanations of changes detected at a systemic level (plasma) based on what is presently known about T cell metabolism. This approach has been used for macrophages, yielding results that have aided in understanding metabolic and immunologic changes, including significant alterations like the proposed Th1/Th2 switch [127,128]. The use of animal models, such as NHP and humanized mice, represents a valuable tool that should facilitate the development of these and other types of studies [112,129,130]. Particularly, the standardization of nutritional factors can help clarify differences observed between studies, which have been attributed to this variable. Notably, while high-throughput techniques identify metabolites of interest, traditional targeted techniques are still needed for validation.

Taken together, the information gathered for the present and other reviews [10,117,126,131] highlights the importance of interconnectivity in the immune response after HIV/*Mtb* coinfection, as we summarize in Figure 2.

## 7. Conclusions

While several questions remain unresolved, the current evidence clearly indicates that coinfection leads to significant changes in the phenotypical and functional profile of T cells, both systemically and locally. These changes are reflected in the depletion of CD4^+^ cell subsets, which dysregulates Th responses and, in turn, affects the functionality of cytotoxic cells. Furthermore, the patterns of T cell differentiation, the rapid use of energy sources, and the general catabolism resulting from the disease lead to severe metabolic changes, with a dependence on alternative energy sources. While some of these changes can be reversed with ART or anti-TB therapy [29,41], others are permanent, and the underlying mechanisms are not fully understood. Future studies should focus on these and other issues, such as the TB-IRIS phenomenon [64], to formulate intervention strategies that can ameliorate or resolve these problems.

The generation of any potential intervention strategies will have to be based on understanding the wide range of effects that are induced by coinfection with HIV and *Mtb*. The interaction between both pathogens is and will continue to be of great importance for the outcome of disease which, in turn, will be highly relevant for public health. Indeed, the treatment of *Mtb* has been proposed as an important part of the prevention and cure of HIV [132]. Thus, a multidimensional approach should be taken into account for the study of the HIV/*Mtb* syndemic, in which any experimental results are interpreted in the light of coinfection, rather than as a result of any individual disease. Likewise, given the magnitude of the issue for public health, the solutions to the HIV/*Mtb* problematic do not solely rely on the biology of the coinfection, but on social and even logistical interventions, and it is likely that a One Health approach would result in more meaningful and successful solutions [133].

## Figures and Tables

**Figure 1 vaccines-12-00901-f001:**
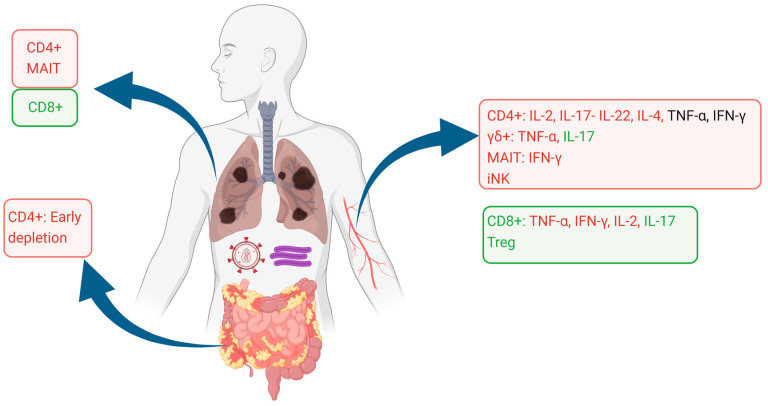
Alterations in T cell phenotypes and functionality of the blood, lung, and gut compartments. Cell subsets are shown as proportionally decreased (red squares) or increased (green squares), as reported in the literature. Cytokine production in each cell phenotype is similarly shown as decreased (red lettering), unaffected (black lettering), or increased (green lettering). Image created using Biorender.com.

**Figure 2 vaccines-12-00901-f002:**
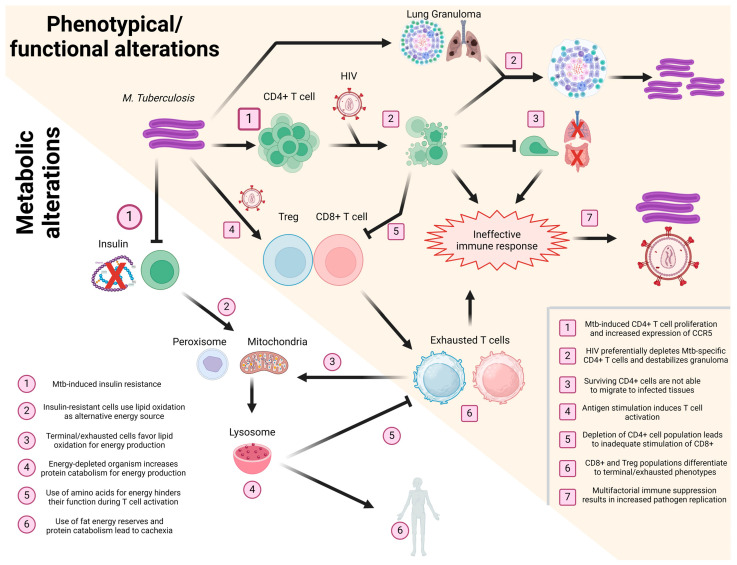
Schematization of the effect of HIV/Mtb coinfection in T cells. Events taking place during coinfection, affecting the phenotype/functionality (yellow background, numbered squares) and metabolism (white background, numbered circles) of T cells, are presented in a sequential manner, with different T cell subsets (CD4^+^, CD8^+^, and Treg) shown in different colors (green, red, and blue, respectively). Image created using Biorender.com.

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
