# Peer review of "T Cell Responses during Human Immunodeficiency Virus/Mycobacterium tuberculosis Coinfection"

_vaccines, 2024, doi:10.3390/vaccines12080901_

Round 1

Reviewer 1 Report

Comments and Suggestions for Authors

1. In the abstract section, it is emphasized that the pattern of differentiation, exhaustion, and transcriptomic changes of T cells during co-infection, as well as the metabolic adaptations necessary for T cells to maintain function, should be discussed. However, this review is limited in space and not fully elaborated during HIV/Mtb co-infection.

2. Paragraphs 251-296, reorder the language in a logical sequence to enhance the readability of the article. For instance, introduce the research background of HIV and Mtb infection separately first, and then present the research focus of HIV/Mtb co-infection.

3. The specific mechanisms mentioned several times in this article, such as the increased expression of CCR5 leading to the increased susceptibility of CD4+ T cells to HIV, can further elaborate on the biological basis behind these mechanisms. Cite relevant experimental evidence or research to support these explanations.

4. Generally, the legend is placed at the bottom of the figure, which appears more aesthetically pleasing and neat. It is recommended to correct it.

5. The numbers in Figure 2 are similar and prone to confusion, and the identification within the figure is unclear and perplexing.

6. Reference 1 is formatted incorrectly, with the year 2023 repeated.

7. Reference 9 has a formatting error. Please rectify it.

8. References 2-7, 12-13, 22, 26, 34, 38-39, 41, 44-45, 51-55, 57-60, 64, 69, 72, 75, 77-79, 81-83, 88-89, 92-95, 98-103, 105-106, 108-109, 111-113, 115, 120, 121-122 lack start and end page numbers.

9. Some of the references are relatively outdated. It is recommended to add references from relatively recent years.

10. The citation reference format needs to be uniform. Recheck and revise it.

Reviewer 2 Report

Comments and Suggestions for Authors

The review on HIV and Mycobacterium tuberculosis (Mtb) co-infection provides a comprehensive synthesis of current literature, effectively highlighting the critical interplay between these pathogens. The discussion on the increased risk of active tuberculosis in individuals with HIV underscores the urgency of addressing this co-infection as a major public health concern.

Your focus on the dual role of T cells as both targets of HIV and mediators of the immune response against Mtb is particularly enlightening. The analysis of how HIV compromises T helper responses, leading to impaired cytotoxic responses, offers a clear mechanism for the observed clinical severity in co-infected patients.

Furthermore, the examination of HIV targeting Mtb-specific cells within granulomas provides valuable insights into the severe disease progression seen in co-infected individuals. This review emphasizes the need for strategies to protect or restore these critical immune cells, making it a significant contribution to the field.

I have a few minor suggestions that could enhance the clarity and comprehensiveness of the review:

Line 37: It would be beneficial to define granuloma structures under the introduction. While these structures are explained elsewhere in the text, providing a few introductory lines with respect to HIV and Mtb can help readers understand how these pathogens coexist in granulomas. Additionally, addressing whether Mtb from infection sites other than the lungs can shed into the lungs would be valuable.

Line 38: The phrase "balance between immune response and pathogen" could be more precisely described as "Mtb pathogenesis" to accurately reflect the content.

Lines 44 and 45: Please mention both new Mtb infections and the activation of latent Mtb to provide a complete picture of the infection dynamics.

Line 48: certain strains induce syncytia in T-cells, forming giant cells within the granuloma structure. It would be helpful to discuss whether these events contribute to the disruption of the granuloma structure.

Line 68: Clarification is needed on how the depletion of CD4 cells by HIV and the increase in CD4 cells by Mtb contribute to the activation of Mtb. This interaction is not entirely clear in the current text.

While the activation of Mtb upon CD4 cell depletion due to HIV infection is well explained throughout the text, the activation of latent HIV in Mtb-infected cells is not covered. Rebound viremia is a major hurdle for HIV cure; does T cell reservoirs such as memory cells infected with HIV and Mtb reverse HIV latency when Mtb activates in these cells? Addressing this aspect could provide a more comprehensive understanding of the co-infection dynamics.

The exploration of T cell differentiation, exhaustion, and transcriptomic changes during co-infection is well-articulated and highlights important areas for future research. The discussion on metabolic adaptations necessary for T cell maintenance and functionality is a crucial addition, as it opens avenues for potential therapeutic interventions aimed at bolstering immune responses.

Overall, this review excellently integrates current findings and presents them in a manner that underscores the interconnectedness of the immune response in HIV/Mtb co-infection. The insights provided are not only academically robust but also hold significant implications for developing targeted treatments and improving clinical outcomes for affected individuals. This work is a valuable contribution to the field and will undoubtedly aid in guiding future research and therapeutic strategies.

Reviewer 3 Report

Comments and Suggestions for Authors

The review provides a comprehensive and informative overview of vaccines, including their development, effectiveness, potential risks, and the interaction between Mtb and the immune system, particularly in the context of HIV coinfection.

Key Areas for Improvement

While the review implies a need for a detailed examination of T cell population alterations due to HIV/Mtb coinfection, it would benefit from explicitly articulating this gap in knowledge. A clear statement on the unique contributions of this review compared to existing literature would enhance its value.

The review thoroughly covers phenotypic alterations in various T cell subsets, including CD4+, CD8+, MAIT, γδ T cells, iNKT cells, and Tregs. This provides a robust understanding of the immune response in this context. However, including more details on

how tissue-specific CD4+ T cell depletion affects TB pathogenesis and treatment in HIV-infected individuals. Clarification on the distinction between absolute numbers and functional capacity of Th1 cells. Further discussion on the significance of impaired Th2 responses in active TB cases and expansion on the implications of metabolic shifts for T cell function and survival would be beneficial.

The discussion section provides a thorough and insightful synthesis of the current state of knowledge on T cell responses during HIV/Mtb coinfection. It highlights the complexity and interconnectivity of immune responses, acknowledges current research limitations, and suggests promising directions for future studies. However, it could be strengthened by providing more specific examples and details, particularly regarding the use of novel technologies and the role of systemic versus local immune modulation.  The discussion effectively synthesizes current knowledge on T cell responses but could be strengthened with more specific examples and detailed analyses on novel technologies used and the role of systemic versus local immune modulation and addressing specific issues related to counterfeit vaccines, discuss the potential impact on public health, details on the mechanisms behind phenotypic changes and their clinical implications, including impacts on disease progression and treatment outcomes.

Minor Comments

  • The term "opportunistic infections, such as Mycobacterium tuberculosis complex (MTBC)" could be confusing as Mtb is a part of MTBC. Clarify the specific role of Mtb within MTBC in the context of coinfection.
  • Simplify sentences for clarity, such as "This depletion is not limited to the bloodstream but occurs preferentially in tissues, including the lungs."
  • Ensure consistent use of terminology, such as "PLWH" (people living with HIV) instead of alternating with terms like "HIV patients."
